# Exploratory Pressure Impregnation Process Using Supercritical CO₂, Co-Solvents, and Multi-Cycle Implementation

Diego Elustondo *[ID], Laura Raymond, Regis Risani, Lloyd Donaldson [ID] and Marie Joo Le Guen [ID]

New Zealand Forest Research Institute (Scion), 49 Sala Street, Private Bag 3020, Rotorua 3046, New Zealand
* Correspondence: diego.elustondo@scionresearch.com

**Abstract:** Supercritical carbon dioxide (scCO₂) is extensively used for extracting chemicals from materials, but the impregnation of materials with chemicals using scCO₂ has received little attention in comparison. To the best of our knowledge, most technologies described in the literature operate by the principle of diffusion, where impregnation yield is limited by solubility. The objective of this exploratory study is to prove the feasibility of an scCO₂ impregnation process that can extract solutes from one material and release them into another material through a single extraction/impregnation stage that can be applied in cycles to increase the yield. The feasibility of the concept was proven in the laboratory using radiata pine bark wax as the solute and radiata pine wood as the impregnated material. Extraction/impregnation tests were performed at temperatures between 40 and 60 °C, pressures between 12 and 16 MPa, and with the addition of ethanol and acetone as co-solvents. The study demonstrated the feasibility of multi-cycle scCO₂ impregnation of wax into wood, where the novelty of the concept is the implementation as traditional pressure impregnation methods.

**Keywords:** supercritical carbon dioxide; co-solvent; radiata pine; bark wax; pressure impregnation





## 1. Introduction

Bark protects trees from the external environment [1]. Bark cells contain relatively large amounts of specialized biomacromolecules such as suberin, extractives, and lipids that play a major role in the hydrophobicity and low permeability of bark [1]. Nature seems to have created a complex material that provides extreme repellence to water, and at the same time has high affinity for hydrophilic wood cells [2]. These materials can be extracted from plants using organic solvents such as heptane, ethanol, diethyl ether, 2-methyl tetrahydrofuran, acetone, toluene, ethyl acetate, propanol, butan-2-ol, dimethyl carbonate and methanol [3].

It was reported that extractives from radiata pine bark were highly hydrophobic and had wax-like film-forming properties [2]. The extractives showed a remarkable affinity for wood as revealed by the high degree of resistance to water penetration and wetting of wax treated wood, even after several wetting/drying cycles [2]. Since bark is a residue from timber manufacturing, there is the research question of whether the same wax that protects living trees from the environment could be used to protect timber products in service.

Impregnation of timber with oils and waxes is common for increasing water repellence and improving outdoor performance properties such as dimensional stability, photostability, and resistance against decay fungi and termites [4]. Impregnation has been performed by pressure treatment with melted wax and wax emulsion [4]. For example, timber was impregnated with melted esterified montan acids, modified plant wax, amid wax, paraffin and montan ester wax at 100 °C and 120 °C [5]. Montan wax in aqueous emulsion was also impregnated in wood by pressure treatment [6].

Pressure impregnation with solvents is a practical method depending on the permeability of timber [7], but it requires drying twice. First to remove water from the fresh timber and then again to remove the solvent. Using supercritical carbon dioxide (scCO₂) as

solvent is an option since $CO_2$ is gas at ambient conditions, thus evaporates spontaneously leaving no solvent behind. $scCO_2$ can also extract hydrophobic waxes from bark [8]; thus, $scCO_2$ could be used for both extraction and impregnating perhaps in a single process. Figure 1 show a conceptual representation of a such hypothetical process.

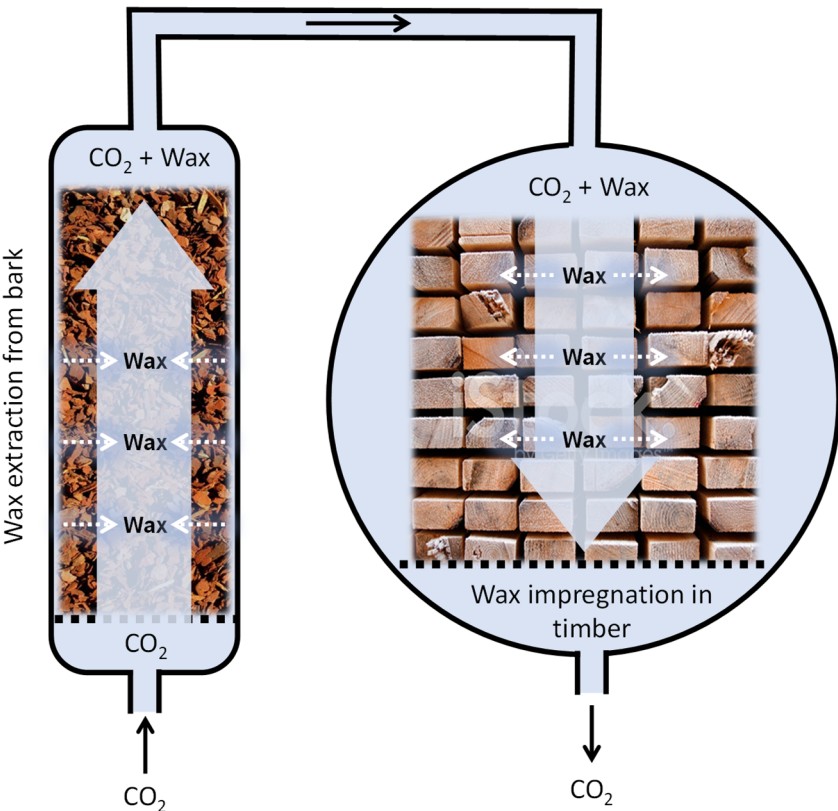

**Figure 1.** Conceptual representation of a hypothetical $scCO_2$ process that extracts wax from bark and impregnates it in timber.

Scion investigated $scCO_2$ impregnation extensively in the past, and the consensus was that it is not practical for timber. Anecdotal evidence suggested that solubility of waxes in $scCO_2$ is very low; thus, impregnation was almost negligible in comparison to what it is required for timber protection. To increase the amount of wax impregnated in timber, the process would need to be implemented in cycles, but this has never been tried before with $scCO_2$ as far as the authors know.

This study revisits the idea of $scCO_2$ impregnation of waxes in wood. First a literature review was carried out to confirm that multi-cycle $scCO_2$ impregnation of wax in wood has not been investigated before. Then, a single-cycle $scCO_2$ impregnation process was tested for different conditions to confirm that wax impregnation is very low in timber. Finally, two options for multi-cycle $scCO_2$ impregnation were proposed and validated at an exploratory level.

## 2. Literature Review

### 2.1. Supercritical $CO_2$ Extraction

$scCO_2$ is extensively used for the extraction of chemical compounds that have low volatility and are susceptible to thermal degradation, especially if there are restrictions for solvents such as in food and pharmaceutical applications [9]. $scCO_2$ extraction is common for natural oils, such as sunflower, tomato, coriander, grape, and peanut seed oils. Solubilities were reported between 2 and 15 mg/g at pressures from 20 to 55 MPa and temperatures from 25 to 50 °C [10].

Other examples are extraction of cottonseed oil at pressures from 35 to 55 MPa and temperatures from 60 to 80 °C [11], and extraction of flaxseed oil at pressures from 21 to 55 MPa and temperatures from 50 to 70 °C [12]. Solubility of flaxseed oil in $scCO_2$ increased from 2.3 mg/g at 21 MPa to 11.3 mg/g at 55 MPa, while corn, soybean, and canola oils reported solubilities between 7.3 and 12 mg/g [12].

$scCO_2$ extraction was also applied to reduce the amount of oil contained in nuts, such as hazelnut, almond, peanut, pecan, and pistachio [13]. Hazelnut oil was extracted at pressures from 15 to 60 MPa and temperatures from 40 to 60 °C. It was found that the solubility increased from 1.0 mg/g at 15 MPa to 28.8 mg/g at 60 MPa [13]. It has been applied to extract natural compounds such as nimbin (which is believed to have insecticidal activity) from neem seeds at 35 °C and 23 MPa [14], alkadienes, carotenoids, β-carotene and γ-linolenic acid from microalgae at temperatures from 40 to 60 °C and pressures up to 35 MPa [15], and lycopene and b-carotene from tomato sauce at temperatures from 35 to 65 °C and pressures from 20 to 30 MPa [16]. The addition of 5% ethanol in $scCO_2$ increased b-carotene and lycopene recovery by, respectively 50 and 55% [16].

The extraction of unsaturated fatty acids from bark with $scCO_2$ was also reported in the literature [8]. It was found that increasing the pressure increases the recovery of wax and sterol esters, while increasing the temperature enhances resin acids solubility. Adding polar organic solvents such as methanol, ethanol and acetone in small amounts achieves higher yields of phenolics, lignans and flavonoids compounds [8]. In summary, $scCO_2$ extraction is a well-known process with many commercial applications that can be used to extract wax from bark.

## 2.2. Supercritical $CO_2$ Impregnation

$scCO_2$ impregnation of solids has been proven feasible, but there is only a handful of commercial applications [17]. These include the impregnation of wood with biocides [18], dyeing of textiles [19], and reactive tanning of leather [20]. All these technologies operate by the principle of diffusion: a material is first pressurized with $scCO_2$ and then a solute is let to diffuse through the $scCO_2$ into the material. To impregnate wood with biocides, $scCO_2$ at temperatures between 40 and 60 °C was first pressurized to 15 MPa and then recirculated through a small mixing vessel to add fungicide [21]. Treatment times were from 1.5 to 5 h.

Dyeing of textiles was implemented in a flow-type cylindrical vessel [22], in which the dye and the fabric were placed in a vessel, and then $scCO_2$ was introduced at a flow rate that assured saturation with the dye. For impregnation of polyester textiles with mango leaf extract, 5% of methanol was added. Temperatures from 35 to 55 °C and pressures from 40 to 50 MPa were used, and treatment times from 15 to 24 h were recommended to maximize impregnation yield [23]. Dying of textiles with $scCO_2$ has been also reported for temperatures from 40 to 160 °C, pressures from 5 to 140 MPa, process times from 5 to 1620 min, and a range of different co-solvents including water, acetone, and alcohol [24].

Impregnation by diffusion tends asymptotically to equilibrium conditions that are determined by the concentration of a solute in a solvent (Darcy's law of diffusion). If the solubility in $scCO_2$ is low, then diffusion is slow and limited to transfer small concentrations of solutes. As far as the authors could confirm, $scCO_2$ impregnation has never been applied as traditional pressure impregnation process. This is a recurring theme in this study because impregnation by diffusion is limited to a maximum determined by the solubility of a solute in $scCO_2$. In this study "traditional pressure impregnation" means that the wax will be dissolved in the $scCO_2$ before it is pushed into the wood by pressure, so that the same cycle can be repeated more than once to increase the impregnation yield. The closest example found in the literature was reactive tanning of leather, in which a tanning solution was deposited on the surface of leather, and then $scCO_2$ was used to push the tanning inside the leather by pressure [17].

## 2.3. Solubility of Fatty Acids in scCO$_2$

A main challenge for implementing scCO$_2$ impregnation as a traditional pressure treatment, is the complex relationship between pressure, temperature, and solubility in scCO$_2$. Selected pressure vs. density isotherms for CO$_2$ are shown in Figure 2, as calculated with an equation of state [25]. The red dot indicates the critical point at approximately 31 °C and 7.4 MPa. Below the critical pressure, there is a range of densities that cannot be realized in normal conditions, where CO$_2$ separates into liquid and gas. The dotted lines in Figure 2 show gas and liquid densities in equilibrium at the same pressure (connected by blue horizontal lines). Above the critical point the CO$_2$ never separates into liquid and gas.

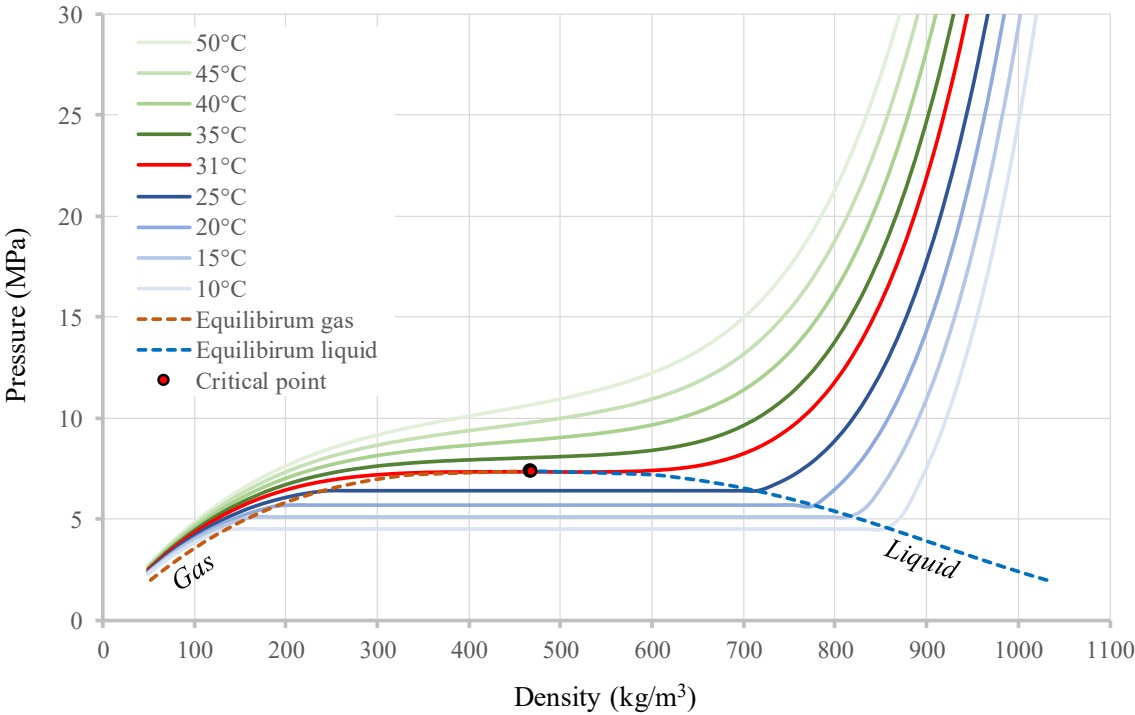

**Figure 2.** Selected pressure vs. density isotherms for carbon dioxide calculated with a published equation of state [25].

Density and temperature are the main parameters affecting the solubility of lipids in scCO$_2$. There are a series of semiempirical models for determining the solubility of solids in supercritical fluids [26]. It is claimed that the "Chrastil model" is the first one [27]. It is based on the theory of chemical association and expressed as shown in Equation (1):

$$ln(S) = A_1 + \frac{A_2}{T} + A_3\,ln(\rho) \tag{1}$$

where $S$ = solubility in a supercritical fluid [kg/m$^3$], $\rho$ = density of the supercritical fluid [kg/m$^3$], $T$ = temperature [K], $A_1$ = empirical parameter (function of the molar mass of the solute), $A_2$ = empirical parameter (function of the enthalpy of solvation and enthalpy of vaporization), and $A_3$ = empirical parameter (association number).

Experimental data of lipids solubility in scCO$_2$ were compiled to calculate the Chrastil model parameters [28]. The compilation included fatty acids, monoglycerides, diglycerides, triglycerides, methyl esters, and methyl esters. For lauric, myristic, palmitic, stearic, oleic and linoleic fatty acids, $A_1$ ranged from 12.0 to $-46.3$, $A_2$ ranged from $-2853$ to $-15{,}890$, and $A_3$ ranged from 5.81 to 9.71. Figure 3 shows an example of solubility calculated using average parameters for the six previous fatty acids ($A_1 = -21.7$, $A_2 = -8211$, $A_3 = 7.48$):

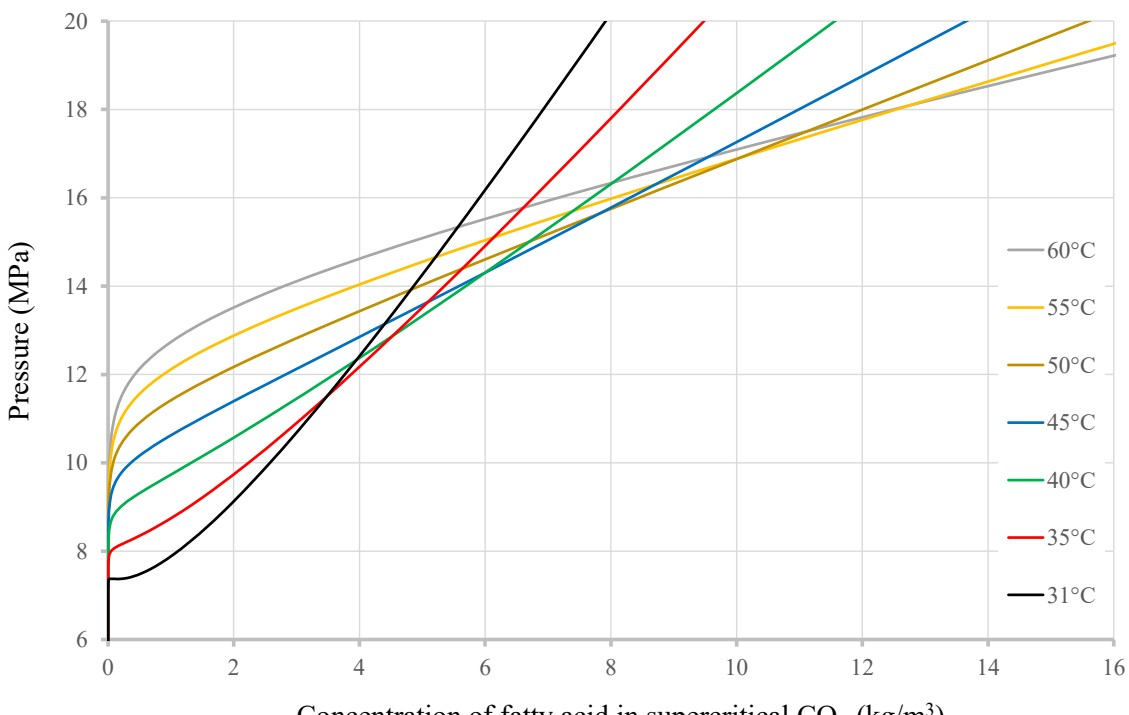

**Figure 3.** Example of solubility in $scCO_2$ calculated using average parameters for six fatty acids ($A_1 = -21.7$, $A_2 = -8211$, $A_3 = 7.48$).

Figure 3 shows that at constant temperature the solubility in $scCO_2$ increases proportionally to pressure, but at constant pressure the effect of temperature is ambiguous. At higher pressures (such as 20 MPa) the solubility increases with temperature, but at lower pressures (such as 10 MPa) the solubility reduces with temperature. The phenomenon is referred as crossover in the scientific literature, and it has been confirmed experimentally [29].

A main challenge to impregnate wood using $scCO_2$ as solvent is that solubility rapidly reduces at low pressures. For example, Figure 3 shows that below 9 MPa the concertation of fatty acids in $scCO_2$ is almost negligible at 40 °C. This means that it is not efficient to dissolve fatty acids in $scCO_2$ at low pressure and then raise the pressure to push the $scCO_2$ into the wood. The $scCO_2$ needs to be pressurized first to dissolve fatty acids at high pressure, and then pushed into the wood. This study proposes a method to implement such process in practice and validates it at an exploratory level.

## 3. Materials and Methods

### 3.1. Experimental Set Up

The process tested in this study is described schematically in Figure 4. There are two pressure vessels at controlled temperature referred as extraction and impregnation vessels. The solute (*Pinus radiata* wax in this study) was placed in the extraction vessel together with a liquid co-solvent when applicable. The wood sample was placed in the impregnation vessel. There were 3 valves controlling the flow of $CO_2$ into (valve 1), between (valve 2) and out of the vessels (valve 3). Valve 1 connected the extraction vessel to a pump (Supercritical Fluid Technologies Inc., Honolulu, HI, USA, SFT-10 CO) that supplied $CO_2$ at controlled flow rate until reaching set-point pressure.

The extraction vessel was an approximately 150 mm long by 30 mm internal diameter stainless steel cylinder (100 mL nominal volume) heated with an external electric jacket. The impregnation vessel was an approximately 159.5 mm long by 14.15 mm internal diameter stainless steel pipe (25 mL nominal volume) submerged in a water bath at controlled temperature. A picture of the actual experimental set-up is shown in Figure 5. The

extraction vessel's output and input valves (7 and 8) are both connected to the top of the vessel, but there is an internal 5 mm diameter pipe inside the extraction vessel that discharges the input $CO_2$ at the bottom.

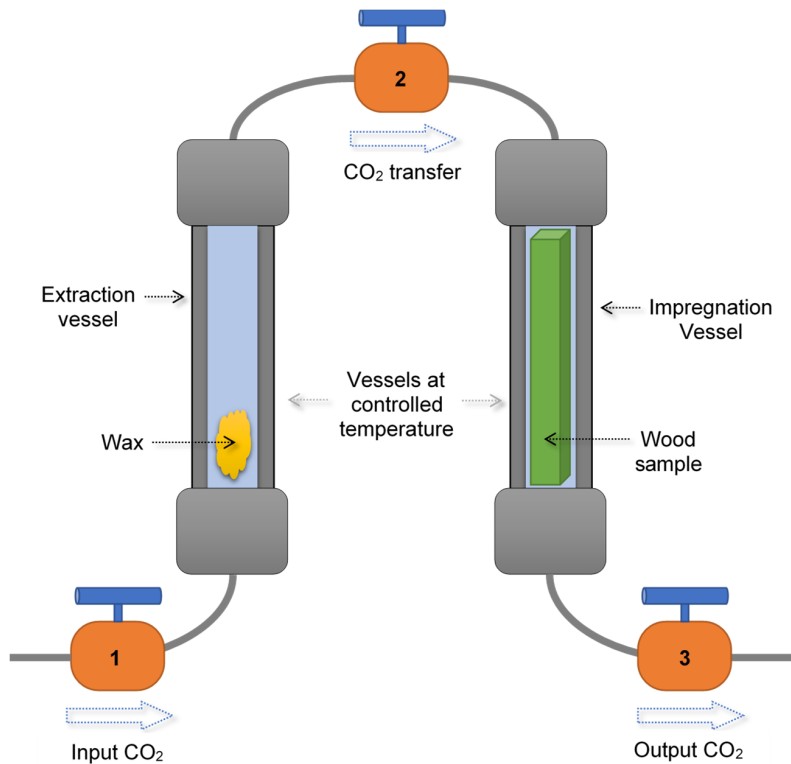

**Figure 4.** Schematic diagram of the $scCO_2$ extraction/impregnation process tested in this study.

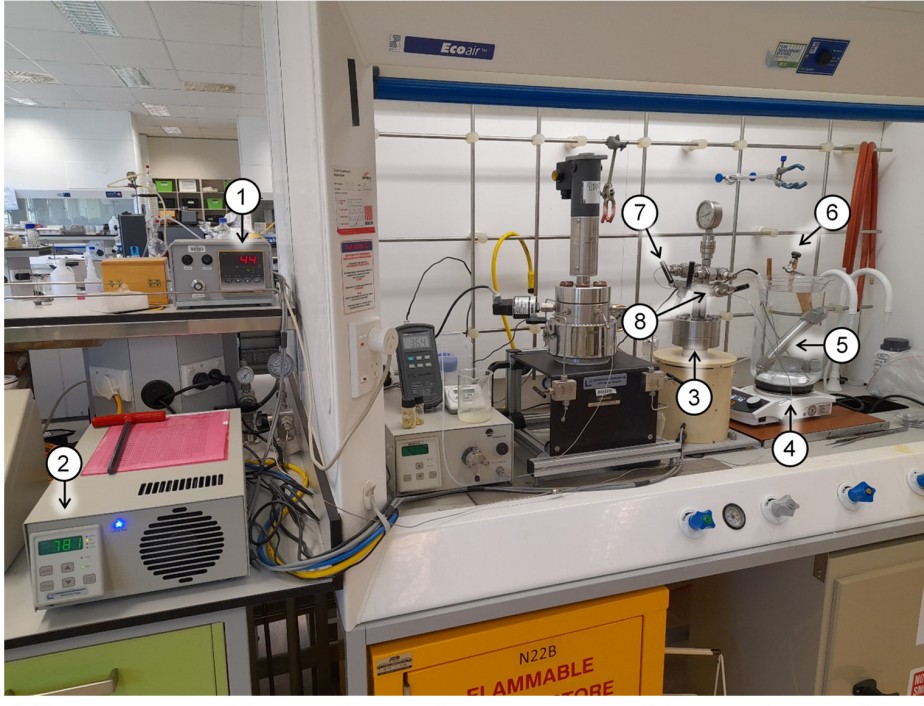

1) Temperature control for extraction vessel
2) $CO_2$ pressure pump
3) Extraction vessel
4) Temperature control for impregnation vessel
5) Impregnation vessel
6) Output $CO_2$ valve
7) $CO_2$ transfer valve
8) Input $CO_2$ valve

**Figure 5.** Picture of the actual experimental set set-up used in this study.

The wood samples were purchased in a local store. The wood product was a 3 m long by 12 mm diameter stick labelled non-treated kiln-dried radiata pine. The sticks were cut into 150 mm long cylinders to fit in the impregnation vessel. Four 3 m long sticks were purchased and cut into four groups of 14 matched wood cylinders for testing. Figure 6 shows an example of a wood sample as it is removed from the impregnation vessel.

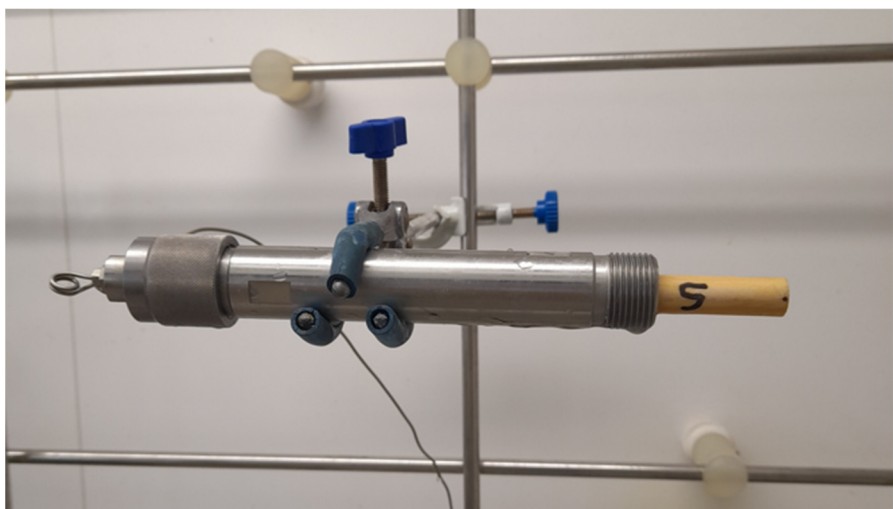

**Figure 6.** Picture of wood sample as it is removed from the impregnation vessel.

*3.2. Process Implementation*

The $scCO_2$ extraction/impregnation process depicted in Figure 4 was implemented according to the following steps:

Step 1: Wax was placed in the 100 mL extraction vessel

Step 2: With valve 1 = open and valve 2 = closed, the extraction vessel was pressurized to the set point pressure by pumping liquid $CO_2$

Step 3: With valve 1 = closed and valve 2 = closed, the extraction vessel was left to stabilize for approximately 1 h at controlled temperature

Step 4: A wood sample was placed in the 25 mL impregnation vessel

Step 5: With valve 1 = closed and valve 3 = closed, valve 2 was opened to connect the two vessels. This step is referred as "pressure drop" in this study

Step 6: With valve 2 = open and valve 3 = closed, valve 1 was opened to repressurise the two vessels to the set point by pumping additional liquid $CO_2$

Step 7: With all valves = closed, the impregnation vessel was let to stabilize for approximately 10 min at controlled temperature

Step 8: With valve 1 = closed and valve 2 = closed, valve 3 was open to depressurize the extraction vessel

Step 9: Depending on the test, a new wood sample was placed in the impregnation vessel (back to step 4), or valve 3 was closed to start another impregnation cycle (back to step 5)

Since the solubility of organic compounds are expected to reduce with density (based on the Chrastil model), some solutes were expected to separate when the supercritical fluid expanded in Step 5 from the extraction to the impregnation vessels (pressure drop). However, solutes that separate inside the impregnation vessel should dissolve later when the supercritical fluid is recompressed to the set point pressure.

The main assumption for the proposed process is that the average density at which $scCO_2$ saturated with wax leaves the extraction vessel (Step 5) is higher than the average density at which $scCO_2$ leaves the impregnation vessel (Step 8). Under this condition, there will be more wax entering the impregnation vessel than leaving the impregnation vessel.

### 3.3. Process Conditions

Two process temperatures were tested in this study: 40 and 60 °C. The first was selected to be slightly higher than the critical $CO_2$ temperature (31.04 °C), and the second was selected to be slightly higher than the wax melting point as determined in this study. The process pressure was selected to ensure that the pressure did not drop below the critical point of an $scCO_2$ + ethanol mixture during Step 5. The pressure drop was estimated theoretically with the $CO_2$ equation of state [25] for a volume expansion from 100 to 125 mL. Theoretical pressures after pressure drops from 12 and 16 MPa are shown in Table 1.

**Table 1.** Theoretical pressures after expanding $scCO_2$ at 12 and 16 MPa from 100 to 125 mL.

| Temperature | Before Expansion | After Expansion |
|:---:|:---:|:---:|
| (°C) | (MPa) | (MPa) |
| 40 | 12 | 9.5 |
| 40 | 16 | 10.1 |
| 60 | 12 | 10.1 |
| 60 | 16 | 13.0 |

The critical pressures for a $CO_2$ + ethanol mixture were estimated from phase diagrams published in the literature [30]. Estimated critical pressures were 8 and 11 MPa at, respectively 40 and 60 °C. This means that pressure in the 60 °C/12 MPa condition would have likely dropped below the critical point for a $CO_2$ + ethanol mixture; thus, it was not tested in this study. The 40 °C/12 MPa, 40 °C/16 MPa, and 60 °C/16 MPa conditions were tested in this exploratory study.

Tests were performed with $scCO_2$ + wax, $scCO_2$ + ethanol, $scCO_2$ + ethanol + wax, and $scCO_2$ + acetone + wax for comparison. All tests were performed in triplicate with matched wood samples. The extraction vessel was filled with wax and co-solvent before the first replicate; thus, subsequent replicates used wax and co-solvent remaining from the previous step. Triplicates for all tests were treated in the same order.

Tests were also designed to have enough wax and co-solvent to complete all cycles without opening the extraction chamber. It was assumed that during impregnation the fluid distributes uniformly between the 100 mL and 25 mL vessels, and then 25 mL was removed during depressurization. For example, 30 g of ethanol should reduce to 12.3 g after four cycles according to the assumption. Experimental tests confirmed that 30 g of ethanol reduced to approximately 12 g after four cycles.

### 3.4. Wax Characterization

It was stated in the introduction the aim of this study was to assess the feasibility of a combined extraction/impregnation $scCO_2$ process that takes wax from pine bark and impregnates it into pine wood. To reduce uncertainty, however, this exploratory study used wax that was already extracted from radiata pine bark using $scCO_2$ extraction equipment at 50 °C and pressures between 10 and 30 MPa.

Chemical composition of the wax was measured as trimethylsilyl ester derivative using gas chromatography–mass spectrometry (GC-MS). A sample was accurately weighed and dissolved in chloroform to give a concentration of around 1 mg/mL. 50 μL of dibromoanthracene (DBA) in pyridine (1 mg/mL) was added as an internal standard. All hydroxyl groups in each compound were derivatised to trimethyl silyl esters by addition of 100 μL of BSTFA:TMCS 9:1 (*v/v*) followed by vortexing and heating at 70 °C for 1 h. Samples were immediately transferred to GC-MS for measuring.

The thermal stability of the wax was measured by thermogravimetric analysis (TGA) with a Discovery TGA (TA instruments, New Castle, DE, USA). About 10 mg of the samples were placed in a platinum sample pan under nitrogen atmosphere (10 mL/min). Temperature was ramped from ambient to 800 °C at 10 °C/min and then maintained for 10 min. Five replicates were measured.

The melting point of the wax was determined with a differential scanning calorimeter (DSC) DSC 214 Polyma (TA instruments, New Castle, DE, USA). For the crude wax, two main state transitions were characterised by DSC, namely, endothermic and exothermic. The endothermic and exothermic transition represents the amount of heat exchanged to, respectively melt and crystalize the sample. No obvious glass transition was observed. Due to peak superimposition related to the numerous components present in the wax, only the total melting and crystallization enthalpiesa were calculated. It was not possible to associate the peaks to specific components.

### 3.5. Oven-Dried Tests

Before and after impregnation, the wood was dried with air at 103 °C for 24 h to determine oven-dry (OD) weight. Since weight gains were expected to be very low, the accuracy of the measurements was assessed at the beginning of the study. The balance was a Mettler Toledo NewClassic ML104/01 (Mettler Toledo Ltd., Greifensee, Switzerland) which provided four decimals on 1.0000 g. All 150 mm long wood samples were oven-dried for 1, 2, 3 and 4 consecutive days to measure OD weight at four different instances. The OD weight was calculated as the average of the four instances, and the measurement error was calculated as the difference between individual measurements and the average.

## 4. Results
### 4.1. Wax Chemical Composition

The main chemical components detected in the wax are reported in Table 2. Chemical composition was measured because the process could have been predicted if Chrastil model parameters were available for the wax components. Unfortunately, stearic acid was the only component for which average Chrastil parameters were available in the reviewed literature [28].

**Table 2.** Chemical composition of principal components detected in the wax.

| Component | Relative % |
|---|---|
| Dehydroabietic acid | 15.83 |
| Docosanoic acid | 11.78 |
| Tetracosanoic acid | 9.17 |
| Tetracosan-1-ol | 6.41 |
| Palustric acid | 6.08 |
| Abietic acid | 5.87 |
| B-Sitosterol | 5.03 |
| Sandaracopimaric acid | 4.68 |
| 1-Docosanol | 3.51 |
| Eicosanoic acid | 3.46 |
| Unknown 1 | 2.85 |
| Isopimaric acid | 2.66 |
| Neoabietic acid | 2.64 |
| Stearic acid (F18:0) | 2.31 |
| Unknown 2 | 2.10 |
| Unknown 3 | 1.65 |
| alpha-Terpineol | 1.22 |
| Unknown 4 | 1.02 |

### 4.2. Wax Thermal Behaviour

The measured degradation ranges and temperatures of melting and crystallization are reported, respectively in Tables 3 and 4. Table 3 shows that the first degradation transition occurs between 62.4 °C and 75.5 °C with a mass loss of 2.8%. To avoid wax degradation, the higher $scCO_2$ temperature tested in this study was 60 °C. Table 4 reports three endothermic and exothermic transitions peaks. Peak 3 was dominant; thus, the melting point was

assumed 49.2 °C. To compare solid and melted wax the $scCO_2$ temperatures tested in this study were 40 °C and 60 °C.

**Table 3.** Degradation transition temperatures and associated mass loss (SD between brackets).

|  | Transition 1 | Transition 2 | Transition 3 |
|---|---|---|---|
| Onset (°C) | 62.4 (4.9) | 244.1 (13.7) | 357.1 (18.1) |
| End of transition (°C) | 75.5 (4.8) | 304.4 (17.5) | 419.3 (16.8) |
| Mass loss (%) | 2.8 (0.7) | 44.4 (1.2) | 48.8 (2.7) |

**Table 4.** Endothermic and exothermic transitions peaks and total enthalpy (SD between brackets).

|  | Peak 1 (°C) | Peak 2 (°C) | Peak 3 (°C) | Total Enthalpy (kJ/kg) |
|---|---|---|---|---|
| Endothermic | 22.93 (0.27) | 29.59 (0.12) | 49.23 (0.34) | 70.19 (1.90) |
| Exothermic | 20.41 (0.28) | 44.70 (0.38) | 52.76 (0.86) | 62.19 (3.52) |

*4.3. Experimental Error*

The results of measuring OD weight at four different instances (1, 2, 3 and 4 days) are summarized in Table 5. The table shows that the average OD weight of all wood samples was 8.02 g, and the experimental error ranged between +0.1% and −0.1%. Errors in the order of +/−0.05% (one standard deviation) should be expected with the implemented experimental set-up.

**Table 5.** Average oven-dried weight (OD), standard deviation (SD) and random measurement error.

| Wood Samples | | Measurement Errors | | |
|---|---|---|---|---|
| OD Weight (g) | SD (g) | SD (%) | Maximum (%) | Minimum (%) |
| 8.02 | 0.35 | 0.05 | 0.10 | −0.09 |

*4.4. Single-Cycle $scCO_2$ Tests*

Table 6 shows a summary of the average weight gain measured in triplicate for single-cycle extraction/impregnation tests, including $scCO_2$ + wax (W), $scCO_2$ + ethanol (Et), $scCO_2$ + ethanol + wax (Et + W), and $scCO_2$ + acetone + wax (Ac + W). The table shows the initial weight of the samples, weight gain immediately after impregnation (in which wood samples still contained solvent and $CO_2$), and weight gain after oven-drying. Tests with only $scCO_2$ + ethanol (without wax) were performed because the wood contained resins and extractives that could be dissolved by ethanol. Tests with only $scCO_2$ where not included because differences could not be discerned from measurement error. Tests with acetone as co-solvent were performed only as reference since ethanol is the preferred solvent in practice.

*4.5. Multi-Cycle $scCO_2$ Tests without Co-Solvent*

To test the possibility of implementing the process in cycles, the 40 °C/16 MPa condition was selected. All wood samples were cut from the same 3 m long radiata pine stick (thus, they were all matched samples). The first set of multi-cycle tests were performed without co-solvent. Approximately 3 g of wax was placed in the extraction chamber and let to dissolve in $scCO_2$ for approximate 1 h. Then, three matched samples were exposed to one, three and six successive extraction/impregnation cycles.

The pressure in the vessels was measured visually with an analogic manometer with resolution of +/−0.5 MPa. The pressure dropped in average from 16 MPa to 10.1 MPa during Step 5, which agreed with the pressure drop estimated theoretically in Table 1. After a total of 10 impregnation cycles, the extraction vessel was opened to confirm that most of the wax was still available in the extraction chamber.

**Table 6.** Summary of average weight gain measured in triplicate for each treatment.

| Test | Process Conditions | Initial Weight [1] | Weight Gain after Impregnation [1] | Weight Gain after Oven-Drying [2] |
|---|---|---|---|---|
| | | g | % | % |
| W-1 | 40 °C/12 MPa | 8.1 (0.3) | 3.7 (1.2) | 0.06 (0.14/−0.04) |
| W-2 | 40 °C/16 MPa | 8.1 (0.2) | 3.5 (0.9) | 0.09 (0.14/0.03) |
| W-3 | 60 °C/16 MPa | 7.9 (0.1) | 0.7 (0.6) | 0.09 (0.12/0.04) |
| Et-1 | 40 °C/12 MPa | 8.1 (0.2) | 11.1 (0.7) | −0.06 (−0.01/−0.16) |
| Et-2 | 40 °C/16 MPa | 8 (0.3) | 12.9 (2.5) | −0.1 (−0.05/−0.17) |
| Et-3 | 60 °C/16 MPa | 8.1 (0.2) | 8.7 (0.5) | −0.11 (−0.07/−0.14) |
| Et + W-1 | 40 °C/12 MPa | 8 (0.3) | 13.2 (2.7) | 0.45 (0.67/0.25) |
| Et + W-2 | 40 °C/16 MPa | 7.9 (0.7) | 16.7 (3.3) | 0.56 (0.81/0.41) |
| Et + W-3 | 60 °C/16 MPa | 7.9 (0.3) | 9.9 (1.2) | 0.32 (0.51/0.22) |
| Ac + W-1 | 40 °C/12 MPa | 7.8 (0.3) | 13.3 (5.5) | 0.37 (0.43/0.31) |
| Ac + W-2 | 40 °C/16 MPa | 8.1 (0.6) | 11.6 (5.2) | 0.34 (0.38/0.3) |
| Ac + W-3 | 60 °C/16 MPa | 8.1 (0.5) | 4.8 (1) | 0.26 (0.32/0.21) |

[1] Standard deviation between brackets. [2] Maximum and minimum values between brackets.

The results are shown in Figure 7. Based on a linear fitting, the wood samples gained approximately 0.064% of their OD weight in wax per cycle. The linear fitting also shows that the first cycle added an additional 0.12% weight gain with respect to the subsequent cycles. Since the extraction vessel was let to stabilize for approximately 1 h before the tests, it was thought initially that this could be the reason for having higher weight gain in the first cycle. However, it was later concluded that this could not be the only reason.

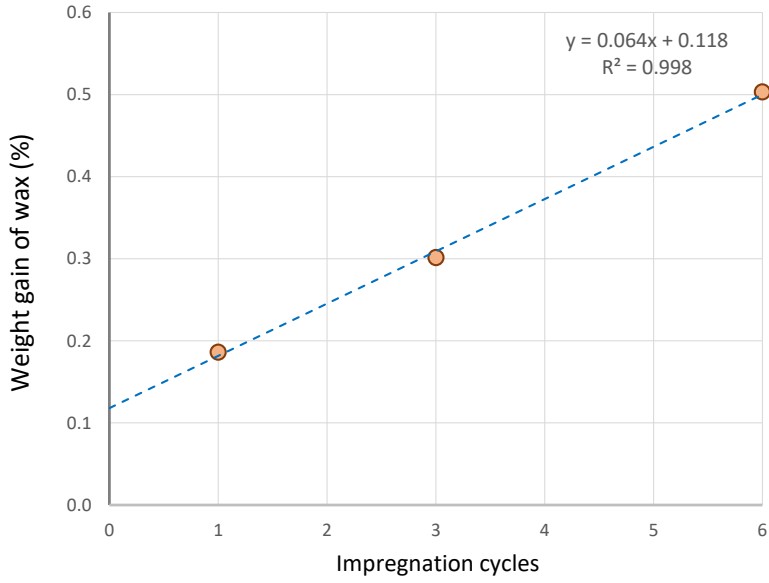

**Figure 7.** Weight gain after 1, 3 and 6 successive extraction/impregnation cycles with supercritical $CO_2$ without co-solvent.

### 4.6. Multi-Cycle scCO$_2$ Tests with Co-Solvent

A second set of multi-cycle tests was performed at 40 °C/16 MPa, with 30 g of ethanol and 1.5 g of wax. Three matched samples were treated, respectively for one, three, and six cycles. Since ethanol is consumed cycle by cycle, then weight gain per cycle could potentially change after one, three and six successive cycles. For this reason, a control sample was treated for one cycle after the sample treated for three cycles, and another control sample was treated for one cycle after the sample treated for six cycles. To minimize the effect of ethanol consumption even further, tests were divided in two groups starting both with 30 g of ethanol and 1.5 g of wax. Results are summarized in Figure 8.

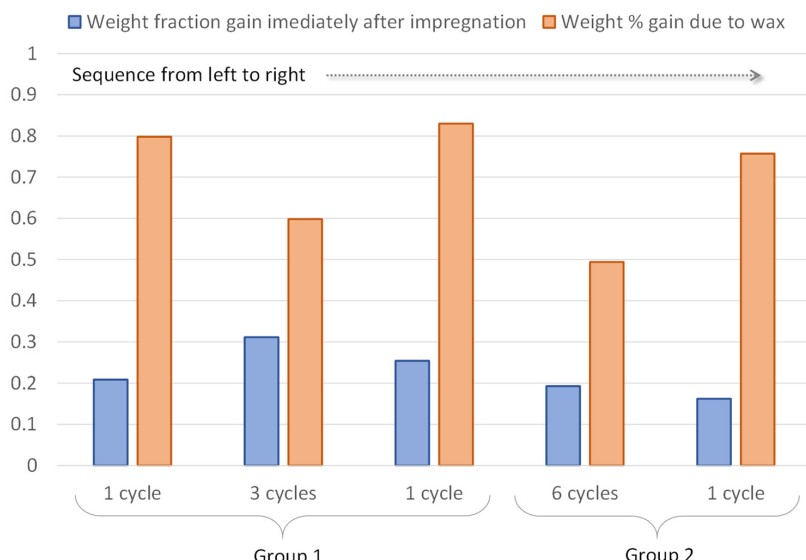

**Figure 8.** Weight gain measured after successive extraction/impregnation cycles. Weight gain is shown as fraction immediately after impregnation and % after oven-drying.

Figure 8 shows that weight gain immediately after impregnation (which still retains ethanol and $CO_2$) and weight gain after oven-drying (which only retains wax) are not considerably affected by the number of cycles. It is apparent from Figure 8 that there is an approximately fixed amount of co-solvent and wax that could be retained inside the wood regardless of the number of cycles. For this reason, a final exploratory test incorporated drying between cycles.

The same five matched samples that were treated in Figure 8, were exposed to three more cycles starting with the same process conditions (40 °C, 16 MPa, 30 g ethanol and 1.5 g wax). The difference was that the samples were oven-dried between cycles. The results are shown in Figure 9. Based on a linear fitting the samples gained approximately 0.30% of their OD weight in wax per cycle. The linear fitting also shows that the first cycle added an additional 0.42% weight gain.

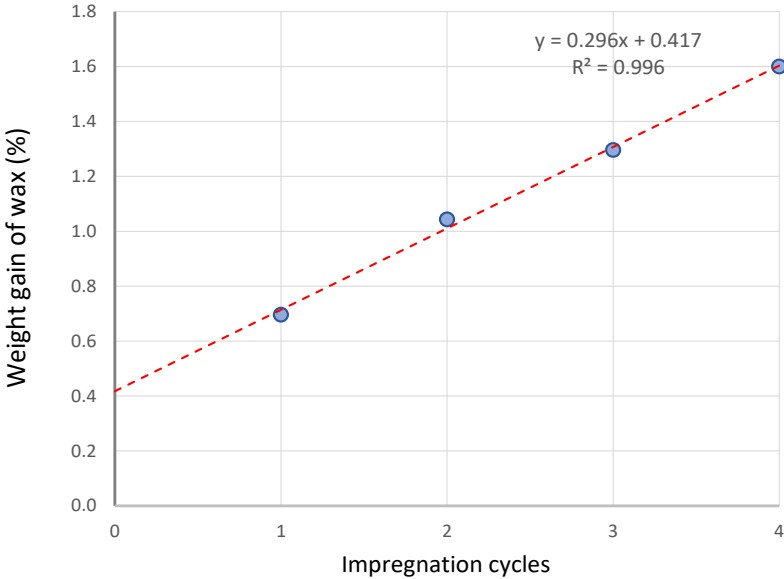

**Figure 9.** Weight gain after successive extraction/impregnation cycles with $CO_2$, ethanol and oven-drying between cycles.

## 5. Discussion

### 5.1. Singe-Cycle scCO₂ Impregnation

The average weight gain reported in Table 6 with $scCO_2$ + wax (without co-solvent) ranged between 0.06% and 0.09%. Those percentages are in the same order of magnitude than the experimental error. However, individual weight gains ranged between 0.14% (maximum) and $-0.04\%$ (minimum), which was clearly shifted to the positive side of the scale (only 1 out of 9 individual weight gains was negative). The averages therefore were treated as empirical evidence of wax impregnation.

A similar trend was observed with $scCO_2$ + ethanol (without wax) but in the opposite direction. The results showed average weight losses between $-0.06\%$ and $-0.11\%$ with a maximum/minimum range between $-0.01\%$ and $-0.17\%$. This was considered empirical evidence that resin and other extractives were removed from the wood during treatment. Treatments with $scCO_2$ + wax + co-solvent (ethanol or acetone) showed weight gains beyond experimental error. The average weight gain was between 0.45% and 0.56% with ethanol and between 0.34% and 0.37% with acetone.

Summing up, single-cycle impregnation tests produced weight gains below 0.1% without co-solvent and below 0.6% with co-solvents. For comparison, impregnation of wood with paraffin wax emulsion was reported in the literature for 300 mm long by 20 mm wide by 20 mm thick loblolly and Scots pine [4]. Paraffin emulsions were diluted to 2% (*w/w*) solid contents with distilled water and impregnated by pressure treatment at 2.0 MPa for 90 min. The study reports average weight gains of 3.5% and 2.6% for, respectively loblolly and Scots pine. This supports the conclusion that the process needs to be implemented in more than one cycle to achieve impregnation levels comparable to traditional pressure treatment.

A conclusion from Table 6 is that wood retained considerably co-solvent and $CO_2$ immediately after treatment. It was found that wood treated without co-solvent retained between 3.5% and 3.7% of their OD weight in $CO_2$, while wood treated with co-solvents retained between 9.9% and 16.7% and between 4.8% and 13.7% of their OD weight in both $CO_2$ and, respectively ethanol and acetone. Further research based on this observation is discussed in Section 5.3 Future research.

### 5.2. Multi-Cycle scCO₂ Impregnation

Figure 7 shows that multi-cycle $scCO_2$ impregnation without co-solvent produced approximately 0.064% weight gain per cycle. If a rounded 3% benchmark is used to compare with traditional pressure treatment, then 45 cycles are required to reach similar impregnation levels. Although exploratory, this suggests that $scCO_2$ impregnation without co-solvent may only be efficient if small amount of wax causes a significant improvement in wood performance.

Figure 8 shows that multi-cycle $scCO_2$ impregnation with co-solvent did not increase weight gain in comparison with a single cycle. One possible explanation is that the solvent separated into an ethanol rich liquid phase during decompression, which was then pushed out from the wood by generation of $CO_2$ gas bubbles. This phenomenon is known as $scCO_2$ dewatering [31].

Evidence of separation is provided by the pressure measured during impregnation. Figure 10 reproduces experimental data of $CO_2$ + ethanol phase equilibrium at 40 °C reported in the literature [30]. Figure 10 also includes the range of pressures measured after pressure drop (Step 5) when vessels were connected, starting with 7 mPa in the first cycle and increasing gradually towards 8 mPa cycle by cycle. Based on the phase equilibrium data for a $CO_2$ + ethanol mixture, at 7 to 8 mPa a liquid phase should separate containing theoretically 50% to 20% ethanol in molar basis.

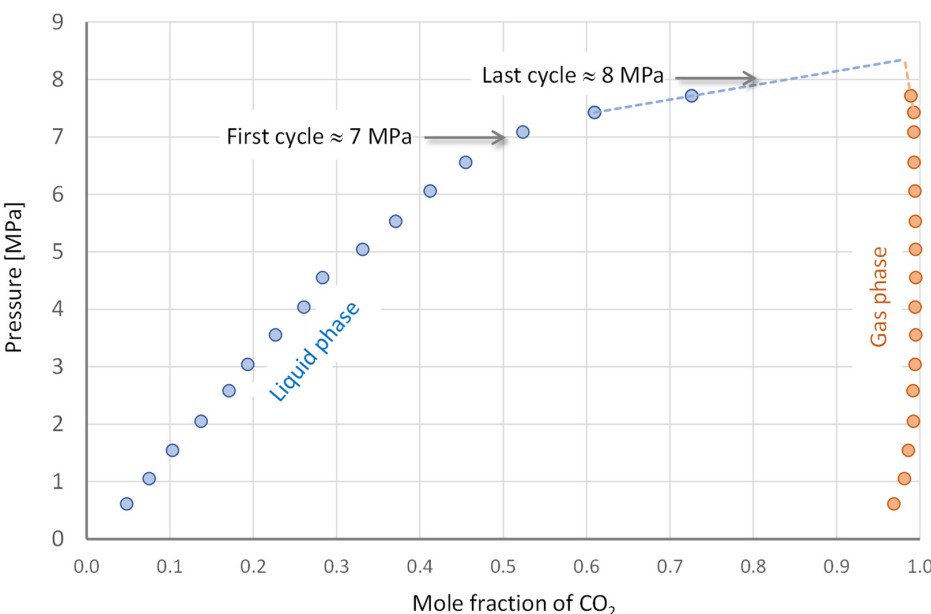

**Figure 10.** Published experimental data of $CO_2$ + ethanol phase equilibrium at 40 °C [30], and pressures measured during multi-cycle impregnation tests.

Because of the dewatering effect during $scCO_2$ decompression (Step 8), each cycle left approximately the same amount of co-solvent and wax inside the wood. To circumvent that limitation the wood was oven-dried between cycles. The assumption was that dewatering removes liquid solvent from the cell lumens but not absorbed solvent from the cell walls. Oven-drying leaves the cell walls available to absorb more solvent in the next cycle.

Figure 9 shows that multi-cycle $scCO_2$ impregnation with co-solvent and drying between cycles added approximately 0.30% weight gain per cycle. If a rounded 3% benchmark is used to compare with traditional pressure treatment, then 9 cycles are required to reach similar impregnation levels. An interesting observation was that the first cycle impregnated more wax than subsequent cycles. Since all cycles in the last exploratory tests started exactly with the same conditions, then it was concluded that untreated wood has higher capacity to retain wax than wood that has been already impregnated with wax.

To support that conclusion treated and untreated samples were inspected with confocal microscope (Leica TCS SP5). Figure 11 shows an image of untreated wood under the microscope, and Figure 12 shows an image of wood treated for four cycles with $scCO_2$ + ethanol and drying between cycles. The green fluorescence in both images is assumed to be caused by oven-dried lignin in cell walls [32], but the brighter yellowish fluorescence mainly appears in the wax impregnated wood. Figure 12 shows ray cells and small cavities that are filled with yellowish fluorescent material, and an image close-up (top left corner) shows yellowish fluorescent material deposited inside cell pits. Since there are only limited pits in cell walls, then this would explain why there is more wax retained in the first cycle.

*5.3. Further Research*

This study demonstrated that it is possible to impregnate wax in wood using $scCO_2$ cycles, but the number of cycles required to achieve levels comparable to traditional pressure impregnation is high at first glance. Further research would require measuring wax penetration into the wood and using fresh bark instead of wax extracted from bark, but a technoeconomic analysis should be performed first based on the preliminary data collected in this study.

It seems apparent (without a technoeconomic analysis) that multi-cycle $scCO_2$ impregnation may only be efficient if small amounts of wax cause a significant improvement in wood performance. In that regard, the finding that wax tends to deposit inside cell wall pits offers opportunities for further research. This should be investigated further because

pits plugged with wax may reduce permeability, and consequently prevent timber from absorbing liquid water in service.

A second interesting observation (that was not anticipated before the study) is that $scCO_2$ impregnation with co-solvents retained a considerably amount of solvent after $CO_2$ decompression. The assumption is that dewatering removes liquid solvent from the cell lumens but not absorbed solvent from the cell walls. This could be further investigated as an opportunity for targeted cell wall modification using a co-solvent that can also react with wood, such as furfuryl alcohol for example [33].

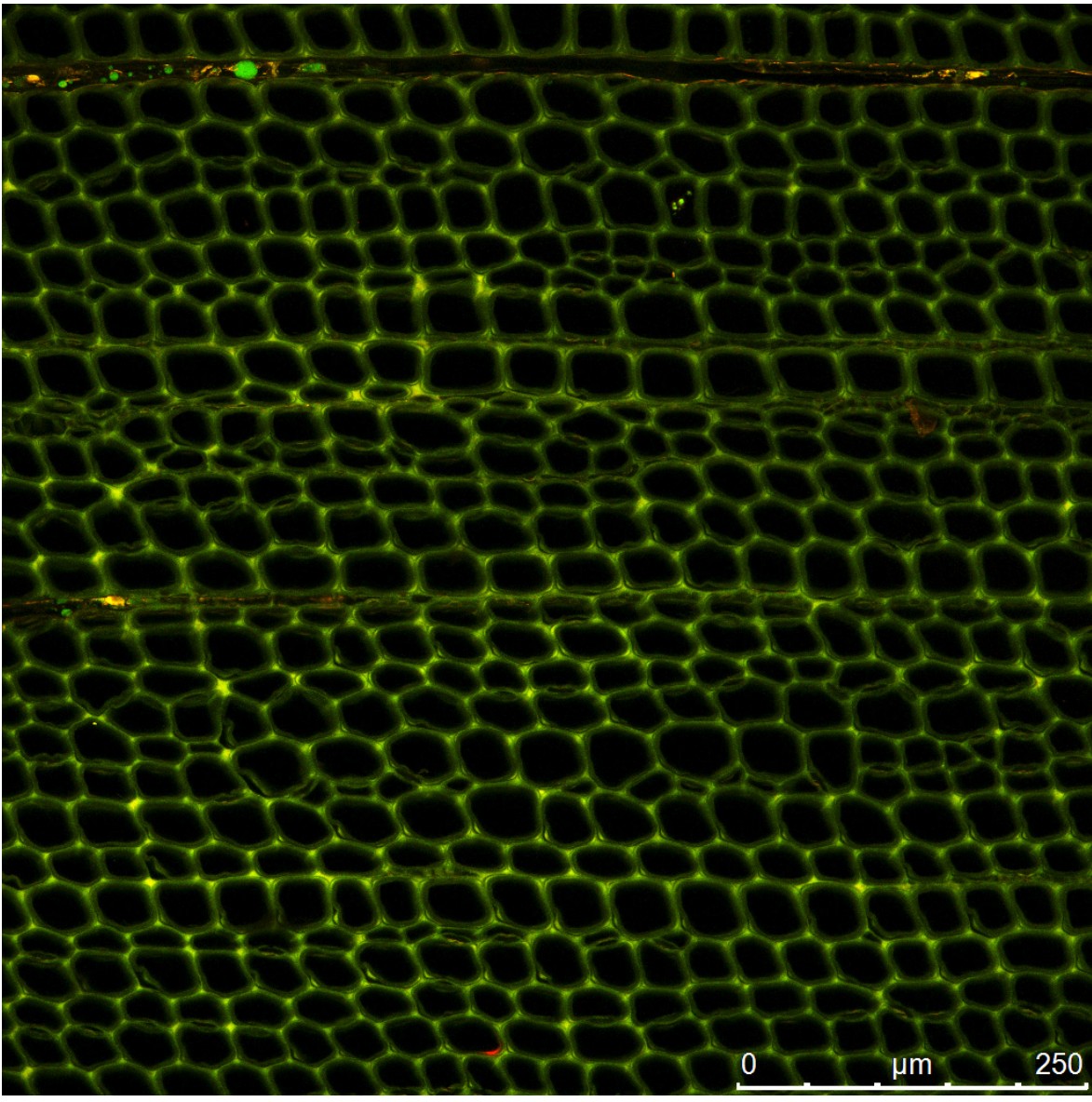

**Figure 11.** Autofluorescence of untreated wood cells under optical microscope.

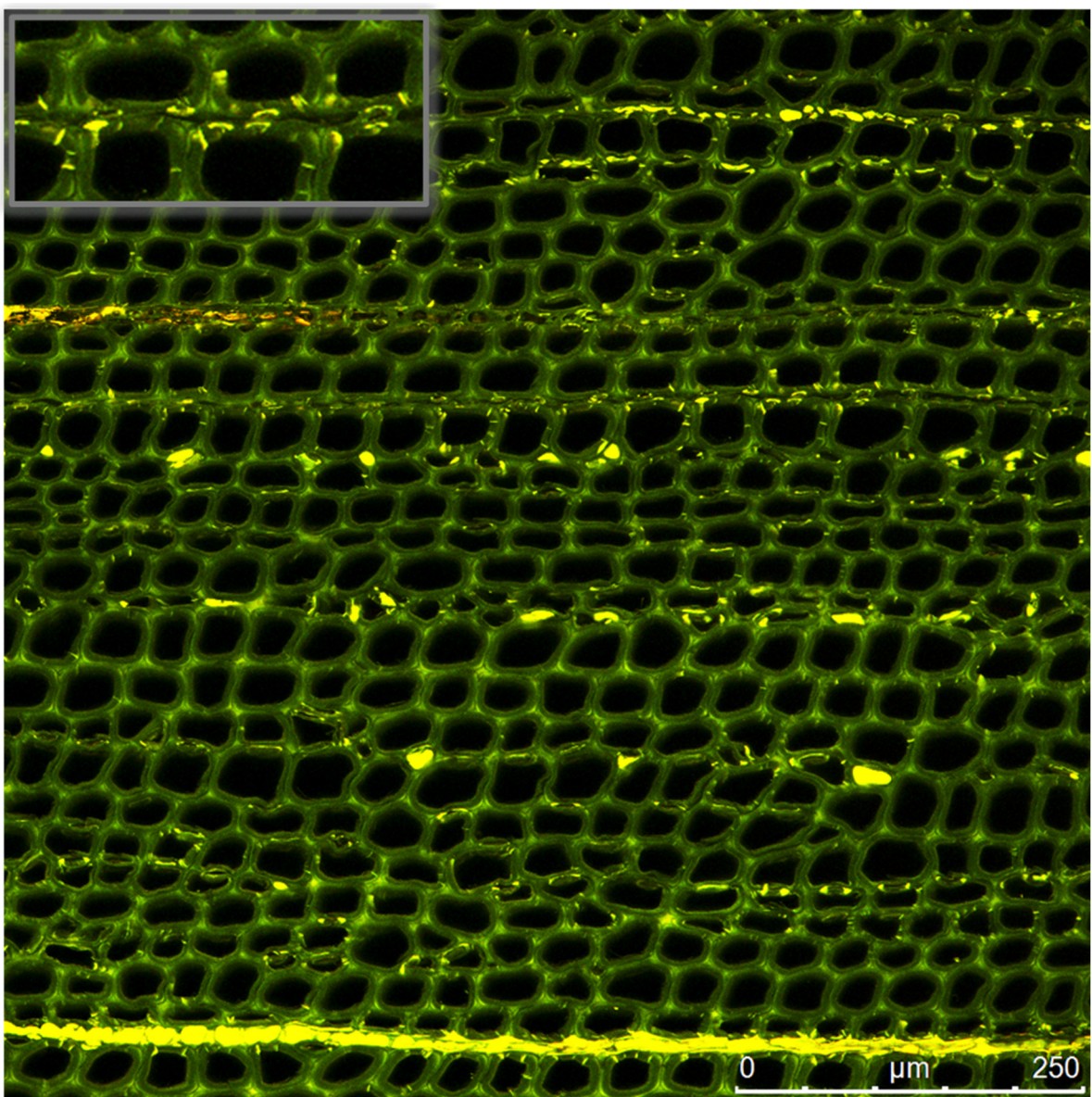

**Figure 12.** Autofluorescence of wax impregnated wood cells under an optical microscope.

## 6. Conclusions

This exploratory study demonstrated that it is possible to impregnate solid wood with bark wax using $scCO_2$ as the solvent. The novelty is that the process was designed to work as traditional pressure impregnation, in which solutes dissolved in a liquid solvent are pushed into the wood by pressure. As far as it was confirmed from the scientific literature, this differs from previously reported $scCO_2$ impregnation methods based on diffusion.

In addition, the study demonstrated at an exploratory level that the proposed process can be implemented in cycles. The results showed that without co-solvent, radiata pine wood samples gained approximately 0.064% of their OD weight in wax per cycle, with the first cycle adding additional 0.12% weight gain.

When ethanol was added as co-solvent, it was found that successive extraction/impregnation cycles did not increase weight gain in comparison with a single cycle, probably because a liquid phase containing ethanol and wax was pushed out of the wood by formation of $CO_2$ bubbles.

The study showed that this limitation can be circumvented by drying the solvent between cycles. After incorporating drying between cycles, samples gain approximately 0.30% of their OD weight in wax per cycle, with the first cycle adding an additional 0.42% weight gain with respect to subsequent cycles.



**Author Contributions:** D.E.: Experimental plan, data analysis and paper writing. L.R. and R.R.: Material properties and chemical analysis. L.D.: Microscopy analysis. M.J.L.G.: Funding and supervision. All authors have read and agreed to the published version of the manuscript.

**Funding:** New Zealand Ministry of Business Innovation and Employment (MBIE) grant number C04X1802 Bark Biorefinery.

**Institutional Review Board Statement:** Not applicable.

**Informed Consent Statement:** Not applicable.

**Data Availability Statement:** The data presented in this study are available on request from the corresponding author.

**Acknowledgments:** Claire Mayer from France's National Research Institute for Agriculture, Food and Environment (INRAe) for her suggestions and feedback.

**Conflicts of Interest:** The authors declare no conflict of interest.

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
