# Peer review of "Exploratory Pressure Impregnation Process Using Supercritical CO2, Co-Solvents, and Multi-Cycle Implementation"

_forests, doi:10.3390/f13122018_

Round 1

Reviewer 1 Report

The manuscript presented a very interesting approach for wood impregnation using supercritical CO2, showing valuable and extended information on procedures and parameters involved. Despite the technology is still a bit far from being scalable to industrial level, the manuscript provides results that are worth publishing. Also, the scientific approach seems very appropriate and well conducted in the lab. It is a pity that authors did not make a good job at the preparation of their submission.

The introduction presents relevant information, but its extension somehow dilutes the purpose of introducing the subject and the relevance of the problem in study. Despite the length, no information on radiata pine wax extraction and the purpose of wood impregnation with wax are presented. Please include them. In addition, the introduction missed a good closing paragraph, paragraph in lines 57 to 61 may be moved to the end of the introduction for that purpose.

Formula in line 132 must be moved to the center of the page

Figure or table referred in lines 147, 152, 167, 109 and 113 are missing

Materials and method section is not excepted with problems.

Reference in line 179 is missing

The paragraph between lines 187 – 190 is incomplete, out of format and it is followed by a figure that is repeated in the next pages (Fig. 5). These are not the type of errors a reviewer must spend his/her time pointing out to the authors, please be more careful with your manuscripts in the future.

References in lines 205 and 213 are missing

Information is missing at the end of line 252.

Subsections showing wax properties and oven dry test must be reformulated as in my opinion table 2, 3,4 and 5 deliver results and hence they should go to results section.

References in lines 288, 296, 307, 308, 309, 321 are missing

Please present your results and discussion in different sections. The extension of the results left few room for discussion, therefore a separate discussion may be more beneficial for the manuscript.

Results shown in Figure 7 are redundant with some results of Table 6. Please choose one of them to present such results.

References in lines 343, 357 380, 383, 402, 415, 416, 424, 431, 440, 444, 446 are missing

Please add a bibliographic citation for your statement in lines 442 to 444.

Different font in lines 479 to 481, please fix that too.

Author Response

Hello,

Thanks very much for your review! I really appreciate it.

I agree with all comments and concerns. They are very good. The manuscript was poorly edited, and I apologize for that. It seems that automatic caption links were lost or corrupted when the file was converted to pdf. In some cases, the caption links repeated figures in the middle of a paragraph or created blank spaces that look as if something was missing. Figures and equations were also displaced to the right of the document while the text was displaced to the left. The edition was awful! Something was wrong in my MS word document. I will make sure that this will not happen again when I upload the reviewed version upon your approval of the proposed changes.

The answers are attached. I grouped the comments and concerns in common themes that could be answered together. Because all reviewers pointed out deficiencies in the “Introduction” and “Results and discussion” sections, I reorganized the manuscript in different sections, thus it was not practical to use “track changes”. Paragraphs that were highlighted with green background indicates information that was not present in the previous version of the manuscript.

I apologize again for the quality. I hope this new version gets closer to the journal standards,

Sincerely, 

Diego

Reviewer 2 Report

The paper studied the pressure impregnation process using supercritical CO2 and the addition of ethanol and acetone as co-solvents with one or multi cycle implementation. The article has a good novelty. Here are some suggestions to improve the quality of your articles.

(1) Since the better impregnation yield is in the process of mixing solvents, it is recommended that the topic include co-solvents instead of only supercritical CO2.

(2) The introduction part 1.1, 1.2, and 1.3. This section is recommended to be streamlined. And This part also needs to added the research background of paraffin-impregnated wood including purpose, function, process, defects, etc.

(3) page 6 is wrong.

(4) Table 6 and Figure 7 weight gain after oven-drying data is duplicated.

(5) In order to improve the impregnation yield, maybe the wood sample should been vacuumed before impregnation.

(6) The reader may pay more attention to the depth of impregnation and want to know if Whether the wood specimen is all impregnated with paraffin, under such high pressure.

Author Response

(The authors gave the same response as above.)

Reviewer 3 Report

First of all, I want to ask the authors to specify the references used (line 109; line 113; line 147; line 152; line 167; line 179; line 205; line 213; line 288; line 296; lines 307-309; line 321; line 329; line 343; line 357; line 380; line 383; line 402; lines 415-416; line 424; line 431; line 440; line 444; line 446).

In my opinion, the Abstract is well prepared, outlining the significance of the study and the main results achieved.

In the Introduction: In my opinion, before point 1.1. "Supercritical CO2 extraction" (line 27) would be good to justify the relevance of the research and the main issues under consideration.

In the text of this part, the considered figures are not presented (this also applies to the rest of the manuscript). Therefore, I ask that this be corrected. The used reference source is missing in Figure 2 (lines 150-151).

In general, in this part, the investigated questions are well outlined, and the relevance of the research is justified. At the end of the section, it would be good to state the purpose of the study.

The second part, "Materials and methods", is well prepared, as the techniques and equipment used are entirely given, and the process parameters are justified based on previous studies. However, due to admitted technical inconsistencies in the cited sources, it is difficult for me to judge their correctness.

The same applies to the "Results and discussion" part.

The Conclusions part is well prepared, reflecting the main results achieved.

Due to the admitted technical inconsistencies, I cannot judge the references' correctness.

Author Response

(The authors gave the same response as above.)

Round 2

Reviewer 1 Report

The authors improved the work considerably. I consider it suitable for publication at its current form.

Reviewer 2 Report

I have no comments and the paper has been improved a lot with the joint efforts of editors, authors and reviewers.

Reviewer 3 Report

The research was very good in the first place. However, as the author admitted, the manuscript was very poorly edited. I am pleased to see that the technical inaccuracies made have been fixed. At this point, in my opinion, the manuscript is excellent.

The respected authors have complied with my recommendations. As a result, the quality of the manuscript is greatly improved, which gives me a reason to recommend accepting the manuscript in its present form.